# Monolithic Saturable Absorber with Gallium Arsenide Nanowires Integrated on the Flexible Substrate for Optical Pulse Generation

**DOI:** 10.3390/mi14091702

**Published:** 2023-08-30

**Authors:** Yifan Zhao, He Yang, Vladislav Khayrudinov, Harri Lipsanen, Xinyang Su, Mei Qi, Baole Lu, Ningfang Song

**Affiliations:** 1School of Instrumentation and Optoelectronic Engineering, Beihang University, Beijing 100191, China; 2Department of Electronics and Nanoengineering, Aalto University, 02150 Espoo, Finland; 3School of Physical Science and Engineering, Beijing Jiaotong University, Beijing 100044, China; 4School of Information Science and Technology, Northwest University, Xi’an 710127, China; 5Institute of Photonics & Photon-Technology, Northwest University, Xi’an 710127, China

**Keywords:** GaAs nanowires, saturable absorber, nonlinear optical modulation, flexible photonics

## Abstract

In this work, we demonstrated a kind of flexibly monolithic saturable absorber (SA) with GaAs nanowires (NWs) on polyimide (PI) plastic substrate for broadband optical modulation at 1.0 and 1.5 µm, separately. The monolithic SA sample was prepared by the metalorganic vapor phase epitaxy (MOVPE) method. The crystal structure and element analysis were examined carefully by high-resolution scanning transmission electron microscopy (HRSTEM) and energy-dispersive X-ray spectroscopy (EDX). We observed a high-density distribution of NWs on the flexible substrate by scanning electron microscopy (SEM). In addition, linear and nonlinear optical properties of the sample were examined by testing the photoluminescence and absorption properties, which showed its potential application as an optical switch due to the pure semiconducting properties. After the characterizations, we experimentally demonstrated this monolithic SA for laser modulation at 1.0 and 1.5 µm, which yielded the minimum optical pulse widths of 1.531 and 6.232 µs, respectively. Our work demonstrated such a kind of monolithic flexible NW substrate-integrated device used for broadband optical modulation, which not only eased the integration process of NWs onto the fiber endface, but also proved the potential of easily integrating with more semiconducting nanomaterials (e.g., graphene, MoS_2_, …) to realize monolithic active flexible photonic systems, such as a microscale phase modulator, delay-line, and so on, paving an easy avenue for the development of both active and flexible photonic devices.

## 1. Introduction

With the continuous development of information networks, miniaturization, integration, high performance, and low power consumption have become inevitable requirements for semiconductor optoelectronic devices [1,2]. Within the past two decades, semiconductor nanowires (NWs) have become a hot research topic in the information field, with their extremely small radial dimensions, easy-to-achieve heterogeneous compatible structural features, and novel electrical and optical properties. Semiconductor NWs exhibit many unique physical properties, such as significant surface [3] and size effects [4], while being free to transport electrons, holes, and photons in the length direction [5]. At the same time, semiconductor NWs also show new effects that are different from those of bulk materials, such as quantum tunneling [6] and quantum interference effects [7]. Therefore, semiconductor NWs are considered ideal building blocks for the development of high-performance photonic and optoelectronic devices, such as solar photovoltaic devices [8,9,10], photodetectors [11,12,13,14,15,16], optical circuits for all-optical nanoprocessors [17], NW waveguides and lasers [18,19,20,21], NW transistors [22], and single-photon detectors [23], etc. Normally, NWs consist of binary or multiple semiconductor compounds, such as III-V and II-IV elements, and the structures have become more abundant with the development of growth and preparation processes, while mostly showing the crystal structures of zincblende, wurtzites, or the combination of both for some defect effects. With the advantages of a direct band gap and high electron mobility, III-V compound semiconductor NWs have broad application prospects in micro- and nano-photonic devices with low power consumption, high speed, and ease of integration [24].

The birth of the world’s first ruby laser in 1960 [25] meant that a high-energy coherent light source was available, allowing direct experimental studies of the nonlinear optical effects of the medium. In 1961, second harmonic generation (SHG) was first found in the lab using a ruby laser by Franken [26]. Thus, various nonlinear optical phenomena can be studied, such as two-photon absorption (TPA) [27], frequency doubling [28], saturable absorption [29,30], four-wave mixing [31], and excited Raman scattering [32], using various nonlinear optical materials, especially with the development of layered materials providing a fruitful material platform to use to engineer the nonlinear optical phenomena. Semiconductor NWs, as a class of one-dimensional nanomaterials, can be reformed as different radial and axial heterostructures [33,34,35], which affects their intrinsic electron mobility and photon transition time, corresponding to the modulation of light–matter response time. Thus, they facilitate the extension of the operating bandwidth to a long wavelength range (e.g., mid-infrared range), which is an ideal material for optical modulation, especially in ultrafast photonic devices.

GaAs is a kind of widely used semiconductor material used for the application of infrared light generation and detection with the main properties of a direct bandgap at 1.42 eV and high electron mobility. GaAs NWs were studied deeply for their application as solar array cells [8] and photodetectors [12,15], and provide great opportunities for integrated photonic and optoelectronic devices. Until now, the majority of GaAs NWs were grown on rigid substrates like GaAs, silicon, quartz, and glass, limiting their potential applications for the highly required integrated and flexible devices, especially the integration with fiber systems. Thus, it is important to study such monolithic devices with easy fabrication methods but wherein they are highly integrated for the development of novel flexible photonic devices.

In this work, we successfully prepared a monolithic device with GaAs NWs integrated on the flexible polyimide (PI) substrate and demonstrated its application for the generation of optical pulses at both 1.0 and 1.5 µm wavelengths. The nonlinear optical absorption properties of NW-integrated flexible devices were investigated by a balanced twin detector method using the laser source at the wavelength of 1560 nm. Then, this monolithic device was applied to the laser cavity to examine the light modulation properties. The minimum pulse width of 6.232 µs at 1.5 µm was obtained under the pump power of 85 mW, whereas the pulse width of 1.0 µm was 1.531 µs at 260 mW. Interestingly, after high-power injection, the monolithic device stood well, implying the properties of a high damage threshold. Our results illustrated a kind of NW-integrated monolithic and flexible saturable absorber (SA) for the potential application as nonlinear optical modulators with high-damage threshold in the field of ultrafast photonic devices.

## 2. Preparation and Characterization of the Sample with GaAs NWs Integrated on the PI (GNiPI) Substrate

### 2.1. Preparation of GNiPI

Self-catalyzed GaAs NWs were grown directly on polyimide tape (Polyonics, Westmoreland, NH, USA, XT-621, 25 µm thick) inside a horizontal flow atmospheric pressure metalorganic vapor phase epitaxy (MOVPE) system. Trimethylgallium (TMGa), tertiarybutylarsine (TBAs), trimethylindium (TMIn), and tertiarybutylphosphine (TBP) were used as precursors. Firstly, the Ga particles were deposited in situ at 480 °C using a TMGa flow of 30.6 μmol/min for 60 s. Next, GaAs NWs growth was started by introducing the TBAs flow (61.20 µmol/min) while maintaining the TMGa flow for 3600 s with the nominal V/III ratio of 2. GaAs NWs were doped in situ by introducing a diethylzinc (DEZn) flow at 0.85 µmol/min during growth. As a last step, GaAs NWs were passivated with an InP shell at 530 °C for 3 s with a nominal V/III ratio of 176 [36]. After growth, the samples were cooled down under TBP flow [37].

### 2.2. Optical Characterizations of GNiPI

To examine the optical properties of our prepared GNiPI, we first studied the optical and mechanical properties of this substrate-integrated GaAs NW. Figure 1a shows the scanning electronic microscopy (SEM: Zeiss Supra 40, Oberkochen, Germany) image of the as-grown GaAs NWs on the PI substrate, which illustrates high-density NW distribution on the substrate suitable for large optical absorption. The inset figure is the optical image of our device, which was used in our lateral pulse laser generation systems. It can be seen that the device was easily bent while the NWs kept standing well on the surface, demonstrating good enough mechanical properties for applications in flexible photonic and optoelectronic devices.

The structural analysis was performed by high-resolution scanning transmission electron microscopy (HRSTEM: JEOL 2200FS, Akishima shi, Japan) and energy-dispersive X-ray spectroscopy (EDX: Oxford, UK, INCA Penta FETx3 spectrometer) to check the crystal structure of our as-grown NW sample. As shown in Figure 1b, the crystalline structure of GaAs was confirmed to be a zinc blende structure under several NW diffraction pattern examinations. Figure 1c shows the wrinkle defects along the zone axis of the GaAs NWs, affecting their optical absorption properties more or less. During the HRSTEM examination, the EDX spectrum was checked, as shown in Figure 1d. It was clearly found that the NWs consisted of Ga and As elements, while Fe and Cu elements originated from the HRSTEM grid and stage.

### 2.3. Linear Optical Properties

Photoluminescence (PL) spectra corresponded to the bandgap of GaAs NWs, with analyses performed under room temperature as in our previous work [37], as shown in Figure 2. The PL spectrum in Figure 2b showed a typical PL peak of undoped GaAs material corresponding to the electron–hole recombination, and the center wavelength was located at 860 nm. Figure 2a shows the PI spectrum of the PI substrate without NWs, verifying that the emission was from the GaAs NWs after extracting the emission properties from the PI substrate. In order to apply it to the near-infrared band, we measured its nonlinear absorption properties.

### 2.4. Nonlinear Optical Response of GNiPI

SA is the core modulation element of passively Q-switched and mode-locked lasers and is characterized by an optical absorption coefficient that decreases with increasing light intensity and eventually saturates. The saturable absorption properties are mainly caused by the Pauli-blocking effect, wherein electrons in the valence band are excited to the conduction band by absorbing incident photons. When the intensity of the incident light is low, most of the photons are absorbed by the material, resulting in low transmittance. When the intensity of the incident beam is relatively high, a large number of electrons are excited into the conduction band, and the energy levels in the conduction band are rapidly occupied and eventually saturated due to the Pauli-blocking effect, which cannot accept more incident electrons. At this point, most of the incident light is not absorbed, resulting in high transmittance. The basic mechanisms for Q switching and mode locking can be easily understood by considering the SA as a passive optical modulator, in which the absorption/loss is intensity dependent. Therefore, it is essential to study the nonlinear absorption properties of the new materials, especially such a GNiPI monolithic SA, including both the nanomaterial and the substrate.

The nonlinear absorption characteristic of GNiPI SA was measured by a balanced twin detector [38], as shown in Figure 3a. The selected ultrafast laser had a pulse width of 100 fs with a central wavelength of 1560 nm and a pulse repetition rate of 80 MHz. A variable optical attenuator (VOA) was used to adjust the intensity of the incident light. A 50:50 optical coupler (OC) was used to distribute the intensity of the probe light and the reference light equally. The optical transmission of the SA under different input light power was then recorded by power meters PM1 and PM2 (Thorlabs PM100D). The obtained experimental results are represented by the blue balls shown in Figure 3b, and the fitted curve is represented by the solid red line. The experimental data were fitted using the following equation [39]:$$T\left(I\right)=1-\frac{\alpha_{s}}{\left(1+\frac{I}{I_{sat}}\right)}-\alpha_{ns}$$

Here, $T\left(I\right)$, $I_{sat}$, $\alpha_{s}$, and $\alpha_{ns}$ denoted the transmittance, saturable intensity, modulation depth, and nonsaturation loss, respectively. $I_{sat}$ was the optical intensity required to achieve half the modulation depth, which determined the characteristics of passively Q-switched and mode-locked lasers. The modulation depth Δ*T* and saturable intensity $I_{sat}$ can be obtained by the theoretical calculation from the experimental data. The theoretical calculation showed the Δ*T* was 5.2% for GNiPI and $I_{sat}$ was 0.86 KW/cm^2^, and then the $\alpha_{ns}$ of 85.2% was obtained from the fitting. Hence, it can be determined that GNiPI SA had high-enough saturable absorption properties for the pulsed laser generation.

## 3. Experimental Setup

The schematic diagram of the proposed Q-switched laser operation based on the GNiPI SA is presented in Figure 4. For 1.5 µm laser operation, a ~1 m long Er-doped fiber (EDF, 12.18 dB/m core absorption at 979 nm, 20.13 dB/m core absorption at 1530 nm) served as the gain medium and was pumped by a 976 nm diode laser (LD: Connet VLSS-976-M-830-1.5-FA). The pump light was injected into the cavity by using a 980/1550 nm wavelength division multiplexer (WDM). A polarization-insensitive isolator (PI-ISO) was placed after the gain medium to ensure the unidirectional operation of the ring cavity. The optical coupler (OC) with a 5% output ratio was selected to extract the light out of the cavity. The GNiPI SA was incorporated into the ring cavity between the PI-ISO and polarization controller (PC). The whole cavity length was ~10 m. The operating wavelengths of WDM, PI-ISO, and OC were all selected at 1550 nm for the EDFL.

## 4. Results and Discussion

### 4.1. 1.5 µm Er-Doped Fiber Laser (EDFL)

The typical experimental results of the Q-switched fiber laser based on GNiPI SA are shown in Figure 5. Figure 5a shows the power output properties under the conditions of continuous wave (CW) and Q switching, respectively. The output power under the CW operating state was higher than that of the Q-switched, irrespective of the input power. It can be easily examined that GNiPI caused a huge loss after insertion into the cavity for the flexible PI materials. Figure 5b illustrates the output spectral properties with the central wavelength of the CW located at 1560.7 nm, whereas the central wavelength of the Q-switching condition stayed at 1530.7 nm. The output spectrum was measured by the spectrometer with the model of Yokogawa 6370. There was a shift of ~30 nm under the insertion of the GNiPI SA caused by the interaction between the laser and the materials, meaning that the huge loss from the materials shifted the center resonance mode in the cavity. The minimum pulse duration of 6.232 µs under the pump power of 85 mW was obtained, illustrated in Figure 5c. The pulse train is placed as an inset to show the steady pulse running state. During the experiment, we found that when the pump power was increased to over 90 mW, the Q-switched operation started to be unstable and the Q-switched pulse even disappeared under the pump power of 125 mW. However, when the pump power was slowly reduced, a stable Q-switched pulse appeared again with the pump power of less than 90 mW, which indicated that the disappearance of the Q-switched phenomena under the high-power case was not due to the damage of the SA, but contributed to the complete bleaching and inability to act as a SA. As illustrated in Figure 5d, the repetition rate increased monotonically from 14.44 kHz to 26.84 kHz with the increase of the pump power from 30 mW to 85 mW. This was the typical passively Q-switched phenomenon: when the pump power increased constantly, the larger gain was to saturate the SA, causing the repetition rate to increase while the pulse duration decreased.

### 4.2. 1.0 µm Yb-Doped Fiber Laser (YDFL)

To test the GNiPI device’s nonlinear configured optical absorption under the wavelength of 1 µm, the same setup as the EDFL ring cavity was employed. A kind of ~1.5 m Yb-doped fiber (YDF, COHERENT SM-YSF-HI-HP, 250 dB/m core absorption at 975 nm, core NA of 0.11) was employed. For the higher power output, an output coupler with a 90/10 ratio was selected to extract the laser emission. The whole cavity length was ~10 m. Correspondingly, the WDM, PI-ISO, and OC operating wavelengths were all operated at 1064 nm for YDFL. Figure 6a illustrates the power output properties under the CW and Q-switching, showing the much higher output power in the CW operating state than that of the Q-switched, irrespective of the input power. The results showed similar high damage threshold properties, implying the potential for the application in the high-energy pulse laser operation. Figure 6b illustrated the output spectrum under the two operation states. It was found that the same phenomenon of resonating wavelength shifted with and without the GNiPI SA. The central wavelength under the CW state was located at 1068.9 nm, while it shifted to 1036.9 nm under the Q-switched operation. A peak shift of ~30 nm was found after the insertion of GNiPI SA for the same reason as the 1.5 µm operation state. Figure 6c showed the single output pulse under the pump power of 260 mW, which was the maximum output power of our LD. The minimum pulse duration of 1.531 µs was obtained under the highest pump power. The corresponding pulse train was inserted as the inset figure to show the steady pulse output. The output pulse width and repetition rate were also recorded to check the pulse properties. As shown in Figure 6d, the pulse width decreased from 3.747 µs to 1.530 µs with the increasing of the pump power from 145 mW to 260 mW. Compared with the pulse properties operated at 1.5 µm, the Q-switched phenomena existed steadily under the maximum pump power, illustrating the device property of high-damage threshold.

### 4.3. Comparison with Other SA Materials

The results of the EDFL and YDFL experiments made it clear that the GNiPI was capable of modulating the laser pulse output performance under the operating wavelength ranging from 1.0 to 1.5 µm. In addition, we calculated the maximum single pulse energy. To compare the pulse generation performance, we include the reported Q-switched performance of other nanomaterials in Table 1. Due to the insertion of the flexible substrate and the scattering effect of light, the loss of our device was relatively high, but it outstood the properties of the high damage threshold, illustrating its potential application in the field of flexible high-energy photonic devices.

## 5. Conclusions

In summary, we prepared a kind of device with GaAs NWs integrated with a flexible plastic substrate and demonstrated it as a kind of monolithic SA for broadband laser pulse generation. Both experimental and theoretical studies were carried out to analyze the nonlinear optical properties of this GNiPI SA. A modulation depth of 5.2% and a saturable intensity of 0.86 KW/cm^2^ at 1.5 µm were achieved after the nonlinear absorption measurement. Furthermore, broadband pulse laser operations at 1.0 and 1.5 µm were demonstrated using GNiPI as a kind of integrated SA. The minimum pulse widths of 1.531 and 6.232 µs were obtained at the wavelengths of 1 and 1.5 µm, respectively. Interestingly, it was found that this GNiPI SA showed a high damage threshold with long-term stability, proving that our GNiPI device holds great potential for applications in nonlinear photonics, especially for applications in the area of flexible high-energy devices.

## Figures and Tables

**Figure 1 micromachines-14-01702-f001:**
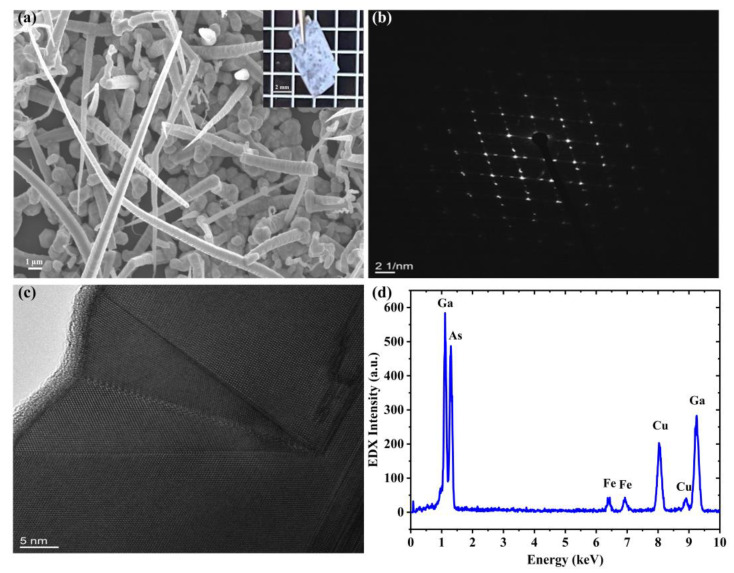
Optical and electron microscopy characterization of GNiPI. (**a**) SEM image of the NWs on the PI substrate surface, and the inset is the optical image of our sample. (**b**) HRSTEM diffraction image. (**c**) HRSTEM defects image. (**d**) EDX pattern.

**Figure 2 micromachines-14-01702-f002:**
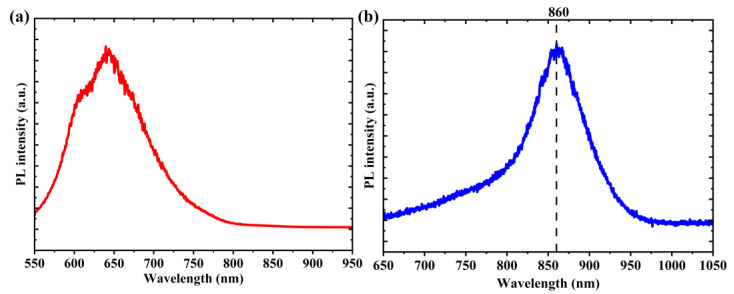
(**a**) Only PI photoluminescence spectrum. (**b**) GNiPI photoluminescence spectrum.

**Figure 3 micromachines-14-01702-f003:**
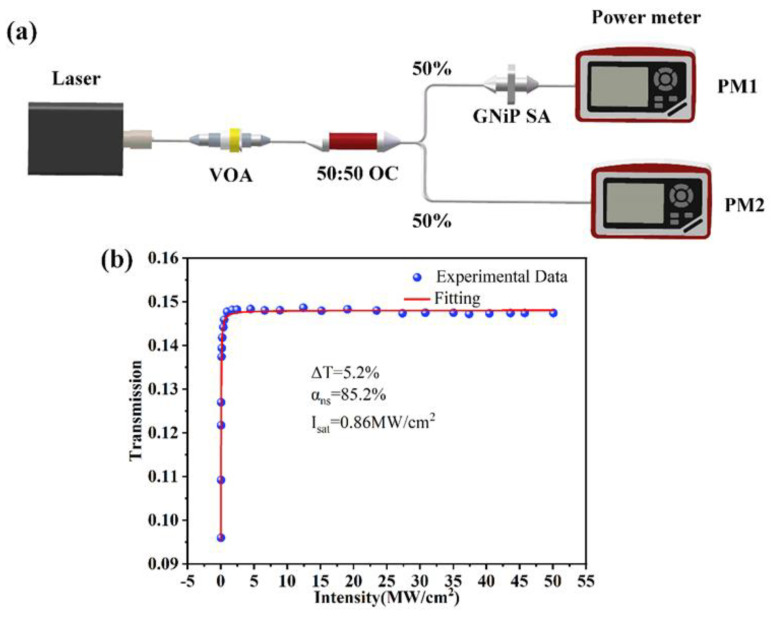
Characterization of the nonlinear optical absorption properties of GNiPI. (**a**) Schematic diagram of GNiPI saturation absorption measurement system. (**b**) Intensity-dependent nonlinear transmittance for GNiPI at 1.5 µm.

**Figure 4 micromachines-14-01702-f004:**
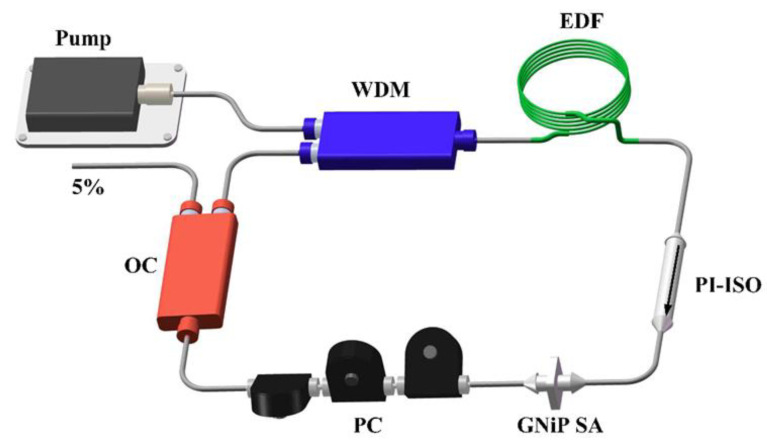
Experimental setup of passively Q-switched EDFL cavity.

**Figure 5 micromachines-14-01702-f005:**
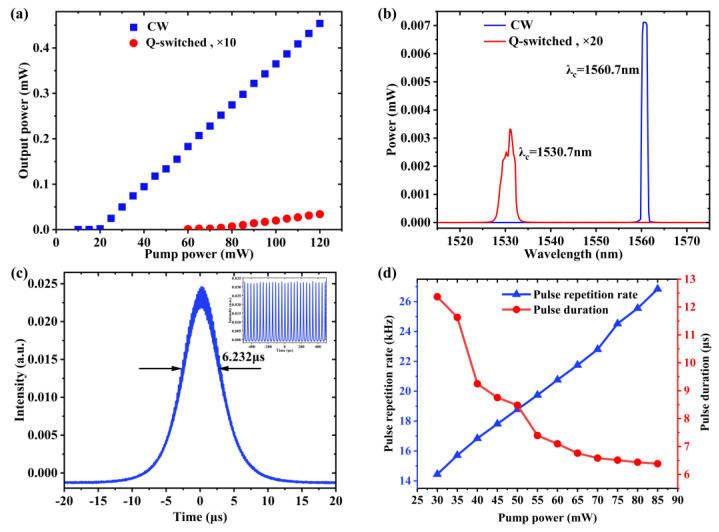
Q-switched laser performance at 1.5 µm based on GNiP SA. (**a**) The variations of output power under both the CW and Q-switching. For ease of presentation, output power from the Q-switched state was scaled up by a factor of 10. (**b**) Output spectrum at 1.5 µm under the two operation states. Similarly, output power from the Q-switched state was scaled up by a factor of 20. (**c**) Single pulse width with a pump power of 85 mW. The inset is the pulse train. (**d**) Q-switched repetition rate and pulse width as a function of pump power.

**Figure 6 micromachines-14-01702-f006:**
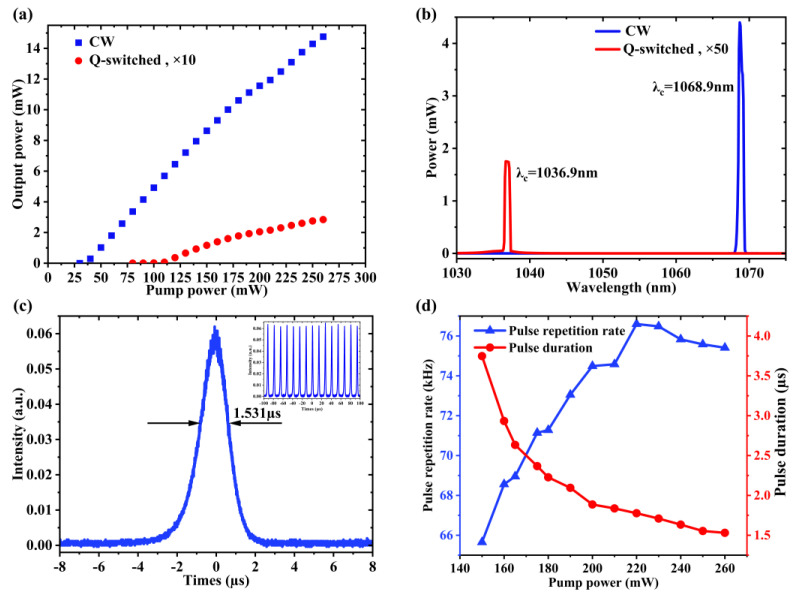
Q-switched laser performance at 1 µm based on GNiPI SA. (**a**) The variations of output power under both the CW and Q-switching. For ease of presentation, output power from the Q-switched state was scaled up by a factor of 10. (**b**) Output spectrum at 1 µm under the two operation states. Similarly, output power from the Q-switched state was scaled up by a factor of 50. (**c**) Single pulse width with a pump power of 260 mW. The inset was the pulse train. (**d**) Q-switched repetition rate and pulse width as a function of pump power.

**Table 1 micromachines-14-01702-t001:** Optical modulation comparison between GNiPI and other nanomaterials.

Wavelength	SAs	Pulse Duration	Repetition Rate/kHz	Single Pulse Energy/nJ	Ref
1.5 μm	CNTs	7.05 μs	~16	14.1	[40]
	CNPs	6.27 μs	73.30	45	[41]
	Graphene	3.7 μs	65.9	16.7	[42]
	γ-graphyne	1.92 μs	132.63	40.35	[43]
	GNiPI	6.232 μs	26.84	0.127	This work
1 μm	CNTs	12.18 μs	24.27	143.5	[44]
	Graphene	~70 ns	257	46	[45]
	InP NWs	462 ns	183	-	[19]
	InAs NWs	411 ns	63	-	[18]
	Bi_2_Te_3_ NWs	303 ns	178.2	1.2 μJ	[46]
	GNiPI	1.531 μs	75.41	3.76	This work

## Data Availability

Data presented in this study are available on request from the corresponding author.

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
