# Peer review of "Monolithic Saturable Absorber with Gallium Arsenide Nanowires Integrated on the Flexible Substrate for Optical Pulse Generation"

_micromachines, 2023, doi:10.3390/mi14091702_

Round 1

Reviewer 1 Report

In this paper, a photoresponsive device based on GaAs NWs grown on a flexible plastic substrate has been successfully prepared, which shows excellent saturable absorption characteristics and small pulse width under photoresponse. There are some small problems, which must be solved before it is considered for publication.

The scale bar in Figure 1(a) is not clear enough.

Figure 1(d) is not clear enough to recognize the elements.

Formula 1 should be centered in the text.

The comparison of GNiPI with other kinds of materials in 4.3 should be more detailed to further illustrate the  strengths and innovations of this study.

Reviewer 2 Report

The paper investigates the monolithic saturable absorber with GaAs nanowires integrated  on the flexible substrate. In the review, the comments are as following.

1. The author should give more detailed information for the equipment used in this studuy.

2. The resolution of Figure 1(d) is very poor. Please give a clearly figure.

3.  The author said that "center wavelength was located at 850 nm for figure 2(b), but I looks the center wavelength is near 900nm, please confirm it. If possible plot a vertical line.

4. Please give more scientific explanation for the center wavelength shift from Figure 2(a) to 2(b).

5. Please give some explanation for the Table 1. Why was this results of this study well than other literature?

moderate 

Round 2

Reviewer 2 Report

All comments have modified and replied. The paper could be accepted as this revised 

All comments have modified and replied. The paper could be accepted as this revised version.